# Spatiotemporal Evolution of Non-Grain Production of Cultivated Land and Its Underlying Factors in China

**DOI:** 10.3390/ijerph19138210

**Published:** 2022-07-05

**Authors:** Zhiyuan Zhu, Zhenzhong Dai, Shilin Li, Yongzhong Feng

**Affiliations:** 1College of Agronomy, Northwest A&F University, Yangling 712100, China; zhuzhiyuan@nwafu.edu.cn (Z.Z.); 2020050047@nwafu.edu.cn (Z.D.); leeshylon@nwafu.edu.cn (S.L.); 2The Research Center of Recycle Agricultural Engineering and Technology of Shaanxi Province, Yangling 712100, China

**Keywords:** China, non-grain production, spatiotemporal evolution, geographical detector

## Abstract

Food security is the foundation of development. We comprehensively characterized the spatiotemporal patterns of non-grain production (NGP) areas in China and elucidated the underlying factors driving NGP. Our objectives were to map NGP on cultivated land (NGPCL) in China, and to quantify its spatiotemporal patterns, to investigate the factors underlying NGP spatial differentiation, and to provide a scientific basis for developing NGP management policies and reference points for protecting cultivated land in other countries. We mapped NGPCL in China from 2000 to 2018 using remote sensing and geographic information system data. The spatiotemporal evolution of the NGP rate (NGPR) was also investigated. The dominant factors driving NGP progression and associated interactions were identified using geographic detectors. From 2000 to 2018, the NGPR gradually decreased from 63.02% to 52.82%. NGPR was high in the west and low in the east, and its spatial differentiation and clustering patterns were statistically significant. Precipitation, temperature, altitude, and soil carbon content were the dominant factors affecting the spatial differentiation in NGPR. The interaction between these factors enhanced the spatial differentiation.

## 1. Introduction

As the world′s most populous country, China has always emphasized food production and the protection of cultivated land. In recent decades, China has experienced rapid economic growth and urbanization, which has been accompanied by patterns that appear contradictory (e.g., in terms of the use of cultivated land) to the emphasis China has traditionally placed on food security [1]. For example, extensive research has been conducted on the conversion of agricultural land into construction land, which has resulted in reduced agricultural output [2,3,4,5]. However, the changes in agricultural land use that have attracted the most interest have been qualitative, not quantitative (i.e., associated with the reduction in agricultural land). Specifically, the “non-grain phenomenon,” which refers to a change in planting structure within cultivated land, has attracted substantial attention from the Chinese government and researchers.

The non-grain phenomenon is widespread in China [6]. A Chinese government document titled “Opinions on preventing non-grain production of cultivated land and stabilizing grain production” emphasizes that high-quality cultivated land should be used for grain production, with an emphasis on the three major grains: rice, wheat, and corn. Farming activities that deviate from this governmental requirement are categorized as non-grain production (NGP) areas. NGP includes the cultivation of cash crops, floriculture, planting trees, and excavating ponds. These activities lead to a major shift away from grain production, resulting in labeling land as non-grain cultivating land. According to the latest statistics from China, cultivated land has reduced by 7.53 × 10^6^ ha over the past 20 years, and the average annual reduction in area is expanding. Clearly, a significant area of cultivated land is being removed from food production, which is attracting the attention of the central Chinese government. Some researchers have concluded that NGP is gradually expanding in China. Currently, the non-grain production rate (NGPR) in China is approximately 27%.

Although NGP enables farmers to diversify their output, thereby potentially increasing their financial gain and promoting China′s rural economic development, NGP has significant drawbacks. The most important aspect of NGP is that it directly reduces the land area used for grain cultivation, which poses a threat to the country′s food security. NGP also threatens local biodiversity, exacerbates non-point source pollution, and increases carbon dioxide emissions.

Previous studies on NGP have been conducted from multiple perspectives. Some have examined the qualitative drivers of NGP, whereas others have focused on environmental and socioeconomic factors. Some studies have even investigated the causes of NGP from the perspective of farmers. However, most of these studies suffer from common limitations. For example, these studies typically examine small study areas, which limits the macroscopic insights that can be gleaned from the present state of NGP in China.

Remote sensing and geographic information systems (GIS) are potent research tools for the large-scale monitoring of cultivated land [7]. With the increasing spatial resolution of remote sensing imagery and continuous technological advancement, remote sensing and GIS can effectively monitor and determine the NGP of cultivated land (NGPCL). However, only a few studies have used remote sensing data to study NGPCL.

To address the lack of national-scale spatial research on NGP, and to provide macro- and large-scale insights for understanding NGP, we comprehensively characterized the spatiotemporal patterns of NGP in China and elucidated the underlying factors that drive these patterns. The specific research objectives were: (1) to map NGPCL in China and quantify associated spatiotemporal patterns; (2) to study the factors underlying the spatial differentiation of NGPR; (3) to provide a scientific basis for developing NGP management policies; and (4) to establish a reference point for cultivated land protection in other countries.

## 2. Materials and Methods

### 2.1. Study Area and Data Acquisition

We investigated 2323 county-level units. These units spanned 31 provinces (limited by the availability of data, not including Taiwan Province, Hong Kong, and the Macau administrative area). Apart from driving factor data, land use data were obtained from the Resource and Environmental Science Data Center of the Chinese Academy of Sciences (https://www.resdc.cn/Default.aspx, accessed on 16 April 2022). In addition, spatial data for wheat, rice, and corn in China during 2000–2018 were obtained from a study by Luo et al. [8]. These data are available at https://data.mendeley.com/datasets/jbs44b2hrk/2, accessed on 16 April 2022. This dataset describes the annual spatial distribution of China′s three most important ration crops at a 1 km resolution. Multiple cropping index data were obtained from a study by Liu et al. [9]. This dataset is a 250 m spatial resolution distribution map of the annual multi-cropping index. Complete data for each county were extracted for geographic detection analysis.

### 2.2. Methodology

#### 2.2.1. Exploratory Spatial Data Analysis

Exploratory spatial data analysis (ESDA) is a collection of methods and techniques for spatial data analysis using spatial correlation measurements. ESDA is crucial for describing and visualizing spatial distribution patterns. In particular, ESDA can reveal spatial agglomeration and elucidate the mechanisms behind spatial interactions between objects [10]. We used ESDA to analyze the spatial distribution pattern of non-grain cultivated land in China from three aspects: global spatial autocorrelation, local spatial autocorrelation, and standard deviation ellipse.

Global spatial autocorrelation was used to determine whether spatial correlations were present between the attribute values of spatially adjacent or disparate area units. The commonly used statistical measures of correlation in spatial statistics are Geary′s C, Moran′s *I*, and Getis′ G, with Moran′s *I* being the most common. Moran′s *I* is represented as a value between −1 and 1. The closer the absolute index value is to 1, the more significant the observed spatial correlation. A negative index indicates a negative spatial correlation, whereas a positive value indicates a positive spatial correlation. When Moran′s *I* is equal to 0, spatial correlation is absent and only randomness exists [11]. According to Moran′s *I*, spatially correlated phenomena are likely to be similar. The following equation was used to calculate the global autocorrelation index:(1)$$I=\sum^{\underset{n}{i=1}}\left(x_{i}-\overset{¯}{x}\right)\sum^{\underset{n}{j=1}}W_{ij}\left(x_{j}-\overset{¯}{x}\right)/\sum^{\underset{n}{i=1}}{\left(x_{i}-\overset{¯}{x}\right)}^{2}\sum^{\underset{n}{i=1}}\sum^{\underset{n}{j=1}}W_{ij}$$
where *n* is the number of spatial grid data points; *x_i_* and *x_j_* are the attribute values of spatial objects at points *i* and *j*, respectively; $\bar{x}$ is the mean of *x_i_* and *x_j_*; and the spatial weight matrix *W_ij_* indicates the strength of the relationship between the *i*th and *j*th points of a spatial object. *W_ij_* can be represented by various parameters, such as area, distance, and reachability.

Global Moran′s *I* indicates the overall correlation between spatial objects, and can identify clustering in the spatial distribution of the object. However, it cannot pinpoint the clustering distribution in space. To pinpoint the clustering distribution, we used the local Moran′s *I* statistic. Unlike the global statistic, local Moran′s *I* measures the spatial correlation between objects in local space. Local Moran′s *I* facilitates the visualization of the spatial clustering of data with a cluster map that delineates and categorizes spatially correlated locations [12]. Local Moran′s *I* is calculated as follows:(2)$$I_{i}=\frac{y_{i}-\overset{¯}{y}}{s^{2}}\overset{n}{\underset{j}{\sum }}w_{ij}\left(y_{i}-\overset{¯}{y}\right)$$
where $\bar{y}$ is the mean; *W_ij_* is the spatial weight matrix; and *s*^2^ represents the discrete variance of *y_i_.*

The standard deviation ellipse is a spatial statistical method that measures the global characteristics of the spatial distribution of geographical elements from multiple perspectives, including concentration, discrete trends, and directional distribution. This method also discerns statistical insights, such as centrality, directionality, and expansion direction deviation. We used the standard deviation ellipse method to explore the spatial evolution of China′s NGP by determining the direction of change of the center of gravity and the dispersion trend. The core parameters of a standard deviation ellipse mainly include the center, azimuth, and major and minor semi-axes.

#### 2.2.2. Geographic Detector (GeoDetector) Model

The GeoDetector model is a statistical method used to investigate the spatial heterogeneity of geographical phenomena for identifying the factors that drive heterogeneity [13,14]. GeoDetector assumes that a study can distinguish multiple subregions. When the sum of the subregion variances is less than the total regional variance, spatial heterogeneity is present. A consistent spatial distribution between two variables indicates a statistical correlation between the variables. The core idea is that if an independent variable strongly influences the dependent variable, the spatial distribution of the independent and dependent variables should be similar. This model includes four sub-detectors: factor, risk, interaction, and ecological detections. This study mainly used factor detection and interaction detection.

The *q* value of each factor is calculated by the factor detector, which quantitatively analyzes the spatial differentiation of each factor, and detects to what extent a certain factor explains spatial differentiation. The following formula was used to calculate factor detection:(3)$$q=1-\frac{\sum_{n=1}^{m}N_{n}\sigma_{n}^{2}}{N\sigma^{2}}$$
where *n* = 1, 2, …, *m*, represents the stratification or partitioning of the independent variable *X* and the dependent variable *Y*; $N_{n}$ and *N* represent the number of units in layer *n* and in the whole area, respectively; and $\sigma_{n}^{2}$ and $\sigma^{2}$ are the dependent variables *Y* in layer *n* and in the whole area, respectively. The variance of the *q* value indicates the explanatory power, and ranges from 0 to 1. The larger the *q* value, the stronger the explanatory power of the independent variable *X* for the dependent variable *Y* and vice versa.

Interactive detection was used to determine whether the interaction of the independent variables *X_m_* and *X_n_* strengthened or weakened the explanation of the dependent variable *Y*, or whether the effects of the independent variables on the dependent variable *Y* are independent. The specific measurement method is to take the driving factors *X*_1_ and *X*_2_ as examples and to calculate the explanatory power *q(X*_1_*)* and *q(X*_2_*)* of the two independent variables to the dependent variable *Y*. Then, the interaction between the two independent variables and the explanatory power *q(X*_1_*∩X*_2_*)* of the dependent variable *Y* are calculated. Finally, the magnitudes of the three calculation results are compared to judge whether the influence of the interaction of two factors on the dependent variable is enhanced or weakened relative to a single factor. The judgment basis is shown in Table 1.

### 2.3. Variable Description

#### 2.3.1. NGPR Measurement

According to the Chinese government document titled “Opinions on preventing the non-grain production of cultivated land and stabilizing grain production,” only rice, wheat, and corn were included in this study. All land cultivation practices other than those for these three crops were defined as NGP. We used NGPR to measure NGP according to the following formula:(4)$$\mathrm{NGPR}=1-\frac{L}{C\;\times \;I}$$
where *L* is the sum of the area of wheat, corn, and rice; *C* is the cultivated land area; and *I* is the multiple cropping index.

#### 2.3.2. Driving Factor Determination

This study refers to previous research findings [5,15,16,17]. Considering the availability and classicality of factor data, we identified the principal selection drivers of NGPR (Table 2). In addition to weather, topography, soil, and socioeconomic variables, 11 representative indicators were selected as driving factors. These factors explain the driving forces at various levels of NGP. Moreover, we used night light indicators to measure the level of urban development [18,19].

## 3. Results

### 3.1. NGPR Measurement and Regional Characteristics

#### 3.1.1. Overall and Regional Characteristics

The national NGPR decreased from 63.02% in 2000 to 52.82% in 2018. However, the NGPR varied across different regions, owing to factors such as natural resource endowment and socioeconomic development. From the perspective of county-level units, the spatial distribution pattern of NGPR characteristics and regional units is relatively consistent. The following points can be observed in Figure 1: (1) NGPR is distributed in a pattern of “high in the west and low in the east.” Among the 2323 county-level units, 1078 have NGPR > 50%, representing 46.4% of the national NGPR. Some hilly and mountainous fields are small, with small per capita areas, which are not conducive to large-scale mechanized farming. Thus, farmers prefer planting crops with high economic benefits in these areas. (2) The spatial NGPR pattern showed a multicenter distribution. A total of 629 county-level units exhibited a non-grain area larger than 5.0 × 10^4^ ha. The spatial patterns of NGPR and non-grain areas were not mirrored across counties. For example, the non-grain area of units with high NGPR was smaller along the southeast coast. In contrast, in the northwest and northeast regions, the NGPR was high, and the non-grain area was widespread.

#### 3.1.2. NGPR Spatial Variability Patterns

Standard deviation ellipse analysis (Figure 2) revealed that the angle of the standard deviation ellipse changed from 56.40° in 2000 to 76.06° in 2018, thus revealing a pattern related to the spatial concentration and evolution of NGPR. This result indicates that NGPR is directed from “northeast to southwest.” The spatial NGPR pattern is also decentralized. Specifically, the long axis of the standard deviation ellipse gradually shortened from 2000 to 2018, whereas the short axis remained largely unchanged. Under the combined action of the long and short axes, the eccentricity gradually decreased. Further, the area of the standard deviation ellipse gradually decreased from 4.33 × 10^8^ ha in 2000 to 4.18 × 10^8^ ha in 2018, thus exhibiting a clear decentralization trend.

### 3.2. Spatial Correlation and Differentiation Patterns of the NGPR

#### 3.2.1. Global Spatial Correlation Characteristics

Significance tests were performed on the global spatial autocorrelation of NGPR. From 2000 to 2018, the global spatial NGPR autocorrelation passed significance tests at the 1% level, indicating that NGPR has a strong spatial correlation. The global Moran′s *I* values for NGPR in 2000, 2005, 2010, 2015, and 2018 were 0.310, 0.274, 0.271, 0.254, and 0.325, respectively. These values indicate a positive spatial autocorrelation. The NGPR distribution was thus spatially clustered, and the degree of spatial clustering increased over time (Table 3).

#### 3.2.2. Local Spatial Differentiation Patterns

A local indicator of spatial association (LISA) cluster distribution map and a Moran scatter plot of NGPR in China′s county areas were calculated for each county. Figure 3 shows 2323 county-level geographic units distributed across four zones: H-H (high-high clustering), L-H (low-high clustering), L-L (low-low clustering), and H-L (high-low clustering). Even at a local scale, the NGPR spatial cluster pattern maintained the same “low in the east and high in the west” distribution as that at the country-wide level. H-H clustering was most strongly associated with arid regions of northern China, the Sichuan Basin, and surrounding areas. The Huanghuaihai Plain, the middle and lower reaches of the Yangtze River, and the Northeast Plain showed L-L clustering, whereas H-L and L-H clustering exhibited sporadic distributions and local fluctuations. Most clustering fell within the first and third quadrants of the Moran scatter plot (Figure 4), indicating the probable dominance of H-H and L-L clustering.

The stable distribution of NGPR with H-H clustering was primarily observed in Xinjiang, the Sichuan Basin, and in parts of the Loess Plateau. From 2000 to 2018, the overall change in NGPR coverage was characterized as “stabilizing in the northwest and expanding to the southwest”. Most of the stable distribution of NGPR with L-L clustering appeared in the Huanghuaihai Plain and parts of the middle and lower reaches of the Yangtze River. The overall coverage from 2000 to 2018 showed that coverage was “stable in the southeast and expanding to the northeast.” The NGPR areas with H-L and L-H clustering were small and sporadic, mainly occurring in peripheral regions adjacent to H-H and L-L clusters. In addition, hot spot analysis indicated the presence of NGPR hot spots in the west, and cold spots in the east. These results corresponded to the LISA cluster distribution map, further supporting the significance of the observed high- and low-value clusters (Figure 5).

### 3.3. Factor Identification of NGPR Spatial Differentiation

#### 3.3.1. Identifying Dominant Factors

In this study, the factor detector method was used to obtain *q*-values for five typical time points in 2000, 2005, 2010, 2015, and 2018. These *q*-values were then ranked (Table 4). The explanatory power of each factor passed the 1% significance level test.

Overall, the factors with the greatest impact on the spatial differentiation of NGPR were annual average precipitation, annual average temperature, elevation, and soil carbon content. Each factor had a *q*-value greater than 0.1.

When examining the dominant factors, interesting patterns emerged. First, the explanatory power of the annual average precipitation, annual average temperature, and elevation decreased over time. Specifically, the *q* value of the annual average precipitation, annual average temperature, and elevation decreased from 0.128, 0.172, and 0.137 in 2000 to 0.062, 0.108, and 0.087 in 2018, respectively. These results reflect a decrease in the influence of natural constraints in shaping NGPR progression. The explanatory power of nighttime lights, distance from highway, distance from railway, and soil carbon content increased between 2000 and 2018. Their *q*-values increased from 0.047, 0.046, 0.023, and 0.105 in 2000 to 0.068, 0.053, 0.035, and 0.132 in 2018, respectively. This result indicates that the influence of socioeconomic development and soil quality factors on the spatial differentiation of NGPR is increasing.

In addition, a stable pattern emerged on ranking the explanatory powers of the most dominant factors. Ranked from highest to lowest, the average annual temperature, elevation, average annual precipitation, soil carbon content, and nighttime lights were the top five factors in 2000. This trend remained mostly unchanged in 2018, with soil carbon content, annual average temperature, elevation, nighttime lights, and average annual precipitation topping the list.

#### 3.3.2. Interaction between Factors

The dominant factors were selected from factor detection, and used to analyze the interaction mechanisms affecting NGPR spatial differentiation to further investigate the changes in the explanatory power of NGPR upon the interaction of different driving factors. Data from the years 2000, 2005, 2010, 2015, and 2018 were analyzed. The results (Table 5) revealed that the factors had a relatively close relationship during the research period, rather than being independent of each other. The *q*-values obtained from the interaction between the driving factors showed different degrees of improvement. The combined effect of two factors typically improved the explanatory power of the NGPR spatial differentiation.

In terms of the types of interaction, 72% of the interactions among the dominant factors showed bivariate enhancement (BE), whereas the rest showed nonlinear enhancement (NE). From the year 2000 to 2018, the explanatory power of each two-factor interaction was different, indicating volatility over time.

Specifically, the explanatory power of the interaction between precipitation and temperature on NGPR decreased from 0.221 in 2000 to 0.143 in 2018. This interaction remained BE, mainly owing to the explanatory power of the decreasing temperature and precipitation during this period. The explanatory power of the interaction between precipitation and elevation initially decreased and then increased. The type of action changed from BE to NE. This change was mainly related to the change in a single factor (elevation). The explanatory power of the interaction between precipitation and soil carbon content decreased and then increased, and the effect type was BE, likely because of the increasing influence of soil carbon content on NGPR spatial differentiation. The explanatory power and action type of the interaction between elevation and nighttime lights showed an unstable change. The explanatory power first decreased and then increased, whereas the action type underwent the following changes: NE → BE → NE → BE. These sporadic changes were mainly caused because the explanatory power of elevation and nighttime light fluctuated over time.

## 4. Discussion

The central government, local governments, and farmers have different goals for cultivated land use [20]. The central government aims to maintain the basic welfare of the populace, local governments aim to pursue local economic growth and improve official performance, and farmers try to maximize their personal economic gains. Thus, the impact of NGP on each group is different. Because planting non-food crops can achieve higher economic benefits and can significantly promote local economic development, some local governments can ignore or even allow NGP behaviors to a certain extent. Farmers are largely driven by economic interests, so they tend to plant cash crops with higher economic returns instead of growing food crops. For the central government, NGP threatens the security of national food supply, and if left unchecked for a long time, NGP may cause substantial costs to manage the resulting economic fluctuations and social unrest. Therefore, NGP caters to the short-term development needs of local governments and farmers, but is not conducive to safeguarding the common interests of the central government and society in the long run [21,22,23,24,25,26].

### 4.1. Impact of NGP on Food Security

NGP first impacts the quality of cultivated land. There are obvious differences in the impact of different types of NGP on the quality of cultivated land. Previous studies have showed that economic crops, such as vegetables and oil crops, have little effect on the ploughing layer. Additionally, if a reasonable crop rotation and effective nutrition practices can be adopted, the quality of cultivated land can be improved. Some fruits can destroy the original soil structure of cultivated land and reduce the quality of cultivated land, but it is relatively easy to restore cultivated land fertility. However, if the cultivated land is used for the production of fast-growing trees, such as poplar, or other types of economic trees with developed root systems, soil degradation, e.g., soil compaction and acidification, often arises because the trees absorb water and fertilizers during long-term planting [27,28,29,30,31,32]. Digging ponds and breeding fish in many areas completely removes the cultivated layer of arable land. In these cases, a large amount of foreign soil is required to rebuild the cultivated layer and gradually restore the land quality. Thus, different NGP types have different effects on the quality of cultivated land, so their role in food security should also be analyzed differently. Moderately supporting NGP behaviors that are beneficial to the quality of cultivated land is beneficial to national food security, whereas long-term laissez-faire NGP behaviors that damage the cultivated layer can seriously threaten national food security.

The second factor is the impact on grain yields in different regions. The main grain-producing areas play an important role in stabilizing national food security. Most studies propose that grain production practices should focus on the main grain-producing areas. Additionally, these studies propose the adoption of strict measures to increase agricultural investment, adjust the structure of grain varieties in a timely manner, and establish a system of preferential interests. It is thus important to prevent NGP behaviors in major grain-producing areas to ensure China′s food security as a whole. In fact, it is unfair to pin the heavy responsibility of ensuring food security on the main grain-producing areas alone. As some major grain-producing areas are economically underdeveloped traditional agricultural areas, these areas bear a heavy responsibility for grain production and pay relatively high costs for arable land protection. The low profit margin of grain crops further affects the economy of the main grain-producing areas. A complete regional compensation system has not been established, which exacerbates the inequitable distribution of grain production responsibilities among different functional areas, thereby threatening the stability of China′s grain production and supply [33,34,35,36,37,38,39,40,41,42,43,44].

### 4.2. Drivers of NGP

The land economic theory states that the essence of land use is the interaction between people and land. Similarly, NGP is the result of interactions between people and land. NGP is affected by nature, economy, society, and institutions. Owing to the comprehensive effect of various factors, the driving factors of NGP are distinct at different scales, but they mainly include subjective factors from farmers themselves, the objective natural environment of cultivated land, and socioeconomic factors [45,46,47,48,49].

Farmers are the most important actors during the utilization of cultivated land. Their behavioral habits, action logic, and psychological expectations directly or indirectly determine the choice of planting behavior, which, in turn, affects the agricultural planting structure. Since China is a large agricultural country with a long history, the effect of the historical and cultural background of farmers cannot be ignored in their planting decisions. With the rapid advancement of urbanization, Chinese farmers are also accelerating their differentiation. Research on the grain-growing logic of new agricultural management entities is conducive to comprehensively and systematically characterize the heterogeneity of Chinese farmers. The natural environment of cultivated land is another important prerequisite for agricultural management choices. Previous studies have found that the natural environment of cultivated land is inseparable from NGP. Scholars typically examine cultivated land quality, geographical location, village type, and topography. A good natural environment for cultivated land is the primary prerequisite for crop cultivation. At the same time, the environment provides the possibility to develop adaptable crops. Strategies to control NGP should be adapted to the land conditions, and differentiated control plans should be proposed according to the natural factors that cause NGP. Socioeconomic development is an important driving force for the transformation of the agricultural structure, and is an important driving factor for NGP. Previous research mainly studied NGP formation from the perspective of economic benefits, farmland transfers, industrial and commercial capital going to the countryside, and grain subsidy policies. With the comprehensive promotion of rural revitalization, a large amount of industrial and commercial capital has been moved to the countryside. However, owing to the profit-seeking nature of industrial and commercial entities, under the guidance of economic interests, the tendency of new agricultural business entities toward NGP is particularly strong. Therefore, driven by rapid urbanization, industrial and commercial capital cannot achieve the original intention of revitalizing the countryside, but instead exacerbate the NGP situation of the planting structure [5,50,51,52,53,54,55].

### 4.3. NGP Control Measures

NGP is affected by the background of rapid urbanization, and is the result of multiple factors, including the farmers interests, the natural cultivated land environment, and socioeconomic factors. Distinct regions exhibit different manifestations. Therefore, identifying and analyzing different NGP types is an important prerequisite for the rational recognition and management of NGP. Future research should clearly distinguish how different non-grain types impact NGP and affect the quality of cultivated land. Further, future studies should propose targeted management and control strategies [56,57,58,59,60].

From the perspective of food security, NGP prevention and control is an important and difficult research topic in China. In general, the foremost problem of NGP control is to solve the problem of the relatively low returns from growing grain. To this end, many studies advocate that grain subsidies should be increased to make grain growing truly profitable for farmers. Fundamentally, the essence of NGP from the perspective of food security is the need to coordinate the competition between short-term economic benefits and long-term food security. Therefore, it is necessary to develop policies at the national level for constructing a “benefit sharing” mechanism for grain-producing areas. The main grain sales, production, and sales balance areas could thus trade grain production indicators for main grain production areas with better arable land resource endowments. This approach can ensure the stability of national grain output and provide moderate support for NGP to help farmers increase their income [61,62,63,64,65,66].

## 5. Conclusions

This study systematically examined the evolution of NGP in China from 2000 to 2018. We specifically investigated the spatial pattern of NGPCL in China and elucidated the key factors driving NGPR. Our results support the following conclusions: (1) From 2000 to 2018, the national NGPR gradually decreased from 63.02% to 52.82%. However, owing to factors such as natural resource endowment and socioeconomic development, NGPR varies greatly in different regions. (2) A significantly positive spatial correlation of NGPR exists in Chinese counties, with the distribution having a geographical clustering effect. The degree of spatial clustering is also increasing. From 2000 to 2018, the overall coverage showed a trend of stable northwest coverage and expansion to the southwest. (3) The dominant factors controlling the spatial differentiation of NGPR are annual average precipitation, annual average temperature, elevation, and soil carbon content. Interactions between these factors improve the explanatory power of NGPR spatial differentiation. The interaction type between the dominant factors is mainly two-factor enhancement, supplemented by nonlinear enhancement.

Overall, this study is a large-scale study on a national scale. On the basis of this study, detailed future studies could be conducted at the plot scale. At the same time, the impact of NGP on the ecosystem and farmers′ grain-growing behavior at the household scale could be investigated.

## Figures and Tables

**Figure 1 ijerph-19-08210-f001:**
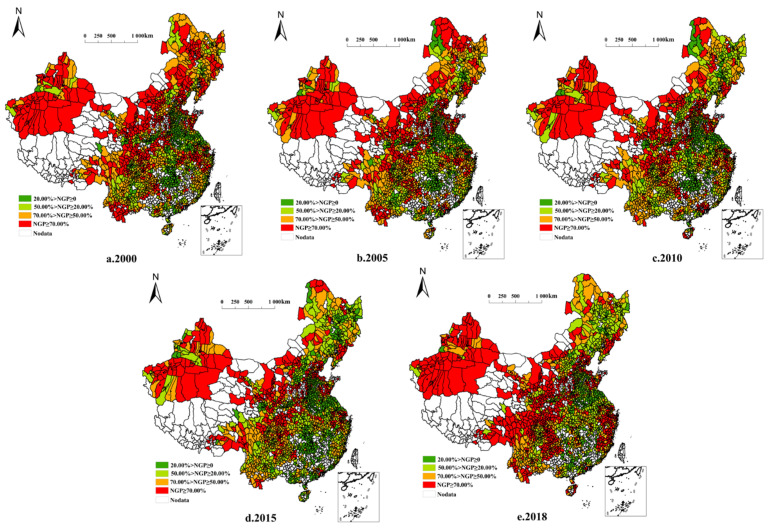
Non-grain production rate of 2323 county-level units in China from 2000 to 2018.

**Figure 2 ijerph-19-08210-f002:**
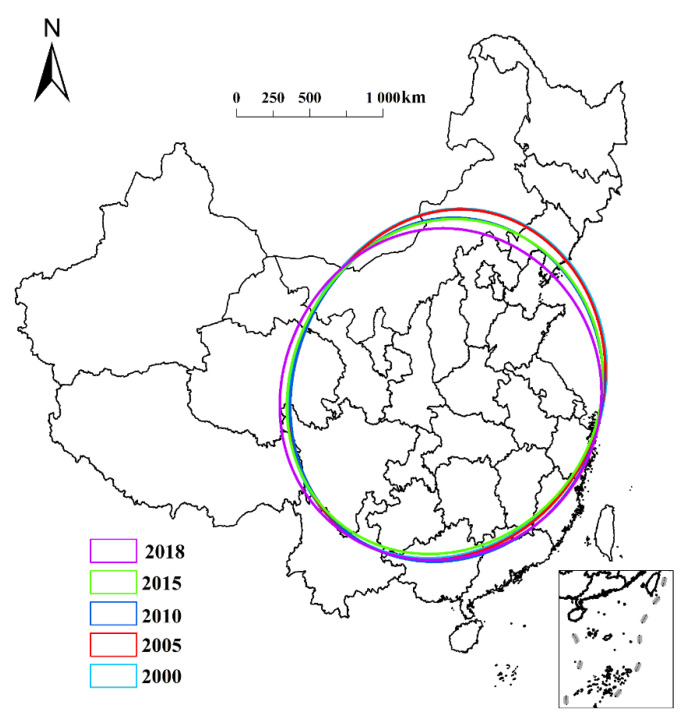
Trend of non-grain production rate in China from 2000 to 2018.

**Figure 3 ijerph-19-08210-f003:**
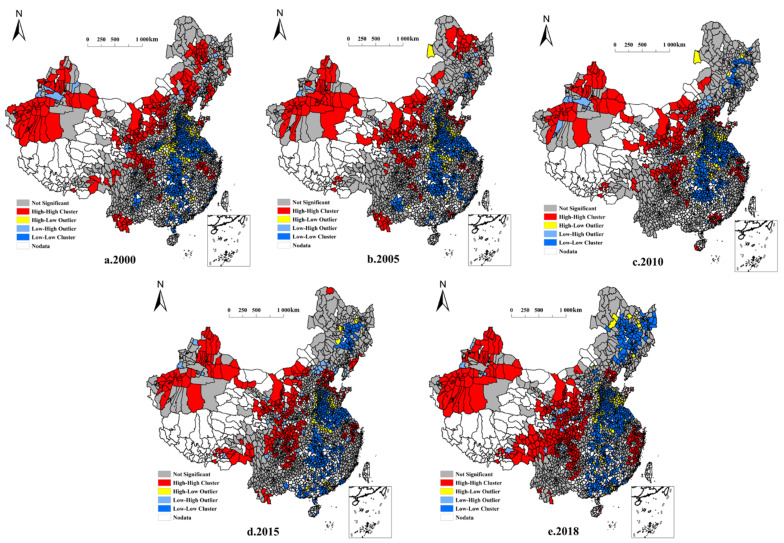
Local indicator of spatial association (LISA) cluster distribution map.

**Figure 4 ijerph-19-08210-f004:**
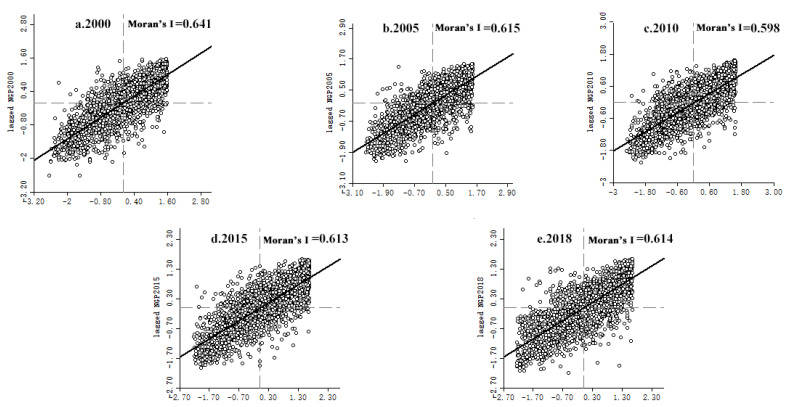
Moran′s I scatter plot of non-grain production rate.

**Figure 5 ijerph-19-08210-f005:**
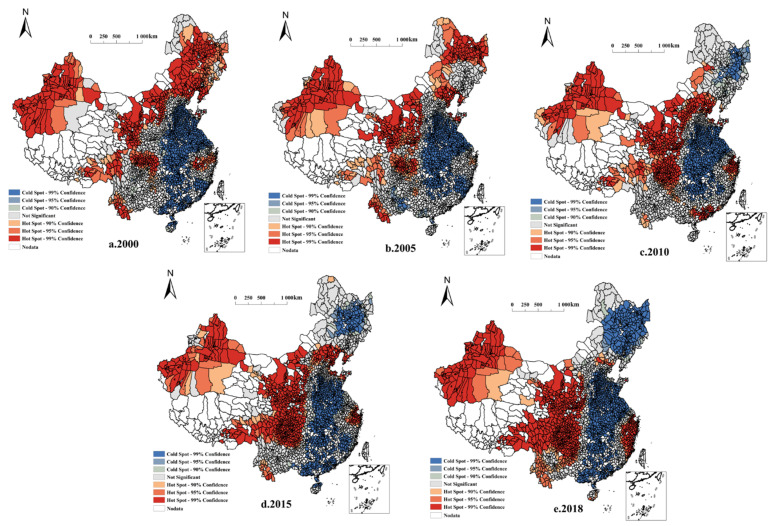
Hot spot analysis of the non-grain production rate.

**Table 1 ijerph-19-08210-t001:** Factor interaction type.

Judgment Basis	Interaction	Code
*q*(*X*_1_∩*X*_2_) < min(*q*(*X*_1_), *q*(*X*_2_))	Nonlinear Weaken	NW
min(*q*(*X*_1_), *q*(*X*_2_)) < *q*(*X*_1_∩*X*_2_) < max(*q*(*X*_1_), *q*(*X*_2_))	Univariate Nonlinear Weaken	UNW
*q*(*X*_1_∩*X*_2_) > max(*q*(*X*_1_), *q*(*X*_2_))	Bivariate Enhance	BE
*q*(*X*_1_∩*X*_2_) = *q*(*X*_1_) + *q*(*X*_2_)	Independent	IN
*q*(*X*_1_∩*X*_2_) > *q*(*X*_1_) + *q*(*X*_2_)	Nonlinear Enhance	NE

**Table 2 ijerph-19-08210-t002:** Driving forces.

	Index	Code	Resolution	Data Sources
Weather factors	Average annual precipitation	*X* _1_	0.1° × 0.1°	University of East Anglia Institute
	Average annual temperature	*X* _2_	0.1° × 0.1°	University of East Anglia Institute
Topography	Elevation	*X* _3_	30 m	Shuttle Radar Topography Mission
	Slope	*X* _4_	30 m	Shuttle Radar Topography Mission
Soil factors	Soil carbon content	*X* _5_	1 km	Harmonized World Soil Database
	Soil organic matter	*X* _6_	1 km	Harmonized World Soil Database
Socioeconomic factors	Population density	*X* _7_	100 m	United Nations Population Density Data
	Night light data	*X* _8_	1 km	NPP/VIIRS night lighting products
	Distance from highway	*X* _9_	/	National Basic Geographic Information Center
	Distance from railway	*X* _10_	/	National Basic Geographic Information Center
	Distance from capital city	*X* _11_	/	National Basic Geographic Information Center

**Table 3 ijerph-19-08210-t003:** Global Moran′s *I* from 2000 to 2018.

	2000	2005	2010	2015	2018
Moran′s *I*	0.310	0.274	0.271	0.254	0.325
Z	90.882	80.496	79.650	74.671	95.314

**Table 4 ijerph-19-08210-t004:** Factor detection results for the spatial differentiation of the non-grain production rate in China.

Factor	2000		2005		2010		2015		2018	
	*q*	Rank	*q*	Rank	*q*	Rank	*q*	Rank	*q*	Rank
*X* _1_	0.128	3	0.100	3	0.086	2	0.072	3	0.062	5
*X* _2_	0.172	1	0.164	1	0.127	1	0.130	1	0.108	2
*X* _3_	0.137	2	0.116	2	0.078	4	0.064	4	0.087	3
*X* _4_	0.030	8	0.034	7	0.027	8	0.041	5	0.035	8
*X* _5_	0.105	4	0.089	4	0.081	3	0.096	2	0.132	1
*X* _6_	0.030	9	0.026	9	0.031	5	0.027	7	0.019	10
*X* _7_	0.025	10	0.021	11	0.011	11	0.023	9	0.012	11
*X* _8_	0.047	5	0.030	8	0.028	6	0.029	6	0.068	4
*X* _9_	0.046	6	0.047	5	0.028	7	0.026	8	0.053	6
*X* _10_	0.023	11	0.023	10	0.014	10	0.016	11	0.035	7
*X* _11_	0.045	7	0.036	6	0.026	9	0.018	10	0.022	9

Note: *X*_1_–*X*_11_ represent average annual precipitation, average annual temperature, elevation, slope, soil carbon content, soil organic matter, population density, night light data, distance from highway, distance from railway, and distance from capital city, respectively.

**Table 5 ijerph-19-08210-t005:** Spatial differentiation interactive detection results for non-grain production rate in China.

Factor Interaction	2000		2005		2010		2015		2018	
	*q*	Type	*q*	Type	*q*	Type	*q*	Type	*q*	Type
*X*_1_∩*X*_2_	0.221	BE	0.180	BE	0.189	BE	0.185	BE	0.143	BE
*X*_1_∩*X*_3_	0.217	BE	0.195	BE	0.148	BE	0.124	BE	0.174	NE
*X*_1_∩*X*_5_	0.198	BE	0.157	BE	0.149	BE	0.127	BE	0.185	BE
*X*_1_∩*X*_8_	0.224	NE	0.172	NE	0.160	NE	0.123	NE	0.172	NE
*X*_1_∩*X*_9_	0.162	NE	0.137	BE	0.109	BE	0.088	BE	0.111	BE
*X*_2_∩*X*_3_	0.243	BE	0.234	BE	0.189	BE	0.185	BE	0.192	BE
*X*_2_∩*X*_5_	0.224	BE	0.209	BE	0.189	BE	0.184	BE	0.201	BE
*X*_2_∩*X*_8_	0.232	NE	0.211	NE	0.174	NE	0.169	NE	0.190	NE
*X*_2_∩*X*_9_	0.183	NE	0.191	NE	0.147	BE	0.144	BE	0.150	BE
*X*_3_∩*X*_5_	0.163	BE	0.139	BE	0.125	BE	0.134	BE	0.175	BE
*X*_3_∩*X*_8_	0.170	NE	0.143	BE	0.101	BE	0.096	NE	0.132	BE
*X*_3_∩*X*_9_	0.142	NE	0.123	NE	0.085	BE	0.068	BE	0.102	BE
*X*_5_∩*X*_8_	0.131	BE	0.105	BE	0.109	BE	0.121	BE	0.161	BE
*X*_5_∩*X*_9_	0.116	NE	0.115	BE	0.093	BE	0.105	BE	0.145	BE
*X*_8_∩*X*_9_	0.091	BE	0.079	NE	0.058	NE	0.057	NE	0.104	BE

## Data Availability

The land use data were obtained from https://www.resdc.cn/Default.aspx, accessed on 20 April 2022. Furthermore, the China 2000–2018 spatial data for wheat, rice, and corn were obtained from https://data.mendeley.com/datasets/jbs44b2hrk/2, accessed on 17 April 2022. The data that support the findings of this study are available from the corresponding author upon reasonable request.

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
