# Peer review of "Spatiotemporal Evolution of Non-Grain Production of Cultivated Land and Its Underlying Factors in China"

_ijerph, 2022, doi:10.3390/ijerph19138210_

Round 1

Reviewer 1 Report

  1. The title is unclear; rewrite it clearly for more clarity.
  2. The abbreviations in the Abstract and all across the draft need to be elaborated the first time they appear. Please avoid uncommon abbreviations in the title.
  3. The abstract language is very poor. Re-write the abstract with concise information delivery of tangible results (L15-29).
  4. The introduction section is very general. Please summarize the rationale, study gap and merits of proposed study with relevant latest references. Hypothesis should be clearly defined (L57-75).
  5. No clarity about the Exploratory spatial data analysis (ESDA)/design (1 or 2 supplementary tables listing all parameters must be added). This will allow the readers to understand the type of Exploratory spatial data analysis (ESDA) and its significance to such studies.
  6. Please use only SI units in the results. Cite only the latest-relevant references only.
  7. Elaborated the Table 1. If possible, draw the correlation/regression equations in Fig. 4.
  8. The discussion needs induction of logical reasoning’s with latest references, which is quite lacking in the draft….Improve it.
  9. Conclusion: please be specific and add future prospects at the end. Currently it looks like repetition of results.
  10. Overall, the english language in manuscript needs extensive improvement. 

Author Response

Response to Reviewer 1 Comments

Point 1:The title is unclear; rewrite it clearly for more clarity.

Response 1: Thanks for your valuable comments. We changed the title to "Spatiotemporal evolution of non-grain production of cultivated land and its underlying factors in China "

Point 2:The abbreviations in the Abstract and all across the draft need to be elaborated the first time they appear. Please avoid uncommon abbreviations in the title.

Response 2: Thanks for your valuable suggestion. We revised the entire text to make sure that the acronyms are defined at their first instance of use.

Point3:The abstract language is very poor. Re-write the abstract with concise information delivery of tangible results (L15-29).

Response 3: Thanks for your valuable advice. We carefully reviewed and polished the language of the abstract.

Point 4:The introduction section is very general. Please summarize the rationale, study gap and merits of proposed study with relevant latest references. Hypothesis should be clearly defined (L57-75).

Response 4: Thanks for your valuable advice. We have added relevant content in the introduction section, better summarized the rationale, identified the research gap, and elaborated on the strengths of the study. We also more clearly defined the study hypothesis.

Point 5:No clarity about the Exploratory spatial data analysis (ESDA)/design (1 or 2 supplementary tables listing all parameters must be added). This will allow the readers to understand the type of Exploratory spatial data analysis (ESDA) and its significance to such studies.

Response 5: Thanks for your valuable advice. To help readers understand the exploratory spatial data analysis we used in this study and its implications, we added a description of the methodology in the Materials and Methods section (Section 2.2.1).

Point 6:Please use only SI units in the results. Cite only the latest-relevant references only.

Response 6: Thanks for your valuable advice. We have carefully reviewed the Results section and to ensure that only SI units are used. We also reviewed the relevant references and cited studies that are as new as possible.

Point 7:Elaborated the Table 1. If possible, draw the correlation/regression equations in Fig. 4.

Response 7: Thanks for your valuable comments. Regarding the description of Table 1, the preceding text contains detailed principle descriptions. Geographic detectors are commonly used geographic research methods, not a new method. We used software and code implementations typical for this analysis, so it is not appropriate to describe the software and code in detail, as this description would be redundant. Regarding the correlation/regression equation drawn in Figure 4, published papers on local spatial correlation indicate that the Moran index does not involve correlation and regression, but instead shows the significance of clustering features. Thus, we cannot add correlation/regression equations to Figure 4.

Point 8:The discussion needs induction of logical reasoning’s with latest references, which is quite lacking in the draft….Improve it.

Response 8: Thank you for your suggestion. We have rewritten the Discussion section to expand on the discussion and to cite related literature. We feel this suggestion improved the quality of the manuscript.

Point 9:Conclusion: please be specific and add future prospects at the end. Currently it looks like repetition of results.

Response 9: Thanks for your valuable advice. We have added future prospects and development directions for this research in the Conclusion section.

Point10:Overall, the english language in manuscript needs extensive improvement.

Response 10: Thanks for your valuable advice. We have had native English speakers revise the entire manuscript. We have attached proof of language editing.

Reviewer 2 Report

Dear appreciated Authors,

the Manuscript: "Spatiotemporal evolution of non-grain production and its underlying factors in China" is very good written, interesting paper and acceptable for publication. While quality of the paper is very good and I suggest minor revision. To improve the quality of the manuscript, I propose the following:

Line 320: "China is a large agricultural country with thousands of years of history; therefore the historical complex of farmers is a factor that cannot be ignored when deciding on planting behavior of farmers.";                                I suggest small changes in sentence in " Since that China is a large agricultural country with thousands of years of history, the historical complex of farmers is a factor that cannot be ignored in planting decisions of farmers."

Line 363: A significant positive spatial correlation of NGPR exists in Chinese counties; the distribution has a geographical clustering effect, and the degree of spatial clustering is increasing.

I would like to suggest small sentence correction in this sentence: 

A significant positive spatial correlation of NGPR in Chinese counties exists, while distribution has a geographical clustering effect and the degree of spatial clustering is increasing. 

Also for formulas in Lines 116, 119, 144 and 173 please delete frame.

Best regards, 

Author Response

Response to Reviewer 2 Comments

Point 1:Line 320: "China is a large agricultural country with thousands of years of history; therefore the historical complex of farmers is a factor that cannot be ignored when deciding on planting behavior of farmers."; I suggest small changes in sentence in " Since that China is a large agricultural country with thousands of years of history, the historical complex of farmers is a factor that cannot be ignored in planting decisions of farmers."

Response 1: Thanks for your valuable advice. We revised this sentence to read, “Since China is a large agricultural country with a long history, the historical complex of farmers cannot be ignored in their planting decisions.”

Point 2:Line 363: A significant positive spatial correlation of NGPR exists in Chinese counties; the distribution has a geographical clustering effect, and the degree of spatial clustering is increasing.I would like to suggest small sentence correction in this sentence: A significant positive spatial correlation of NGPR in Chinese counties exists, while distribution has a geographical clustering effect and the degree of spatial clustering is increasing.

 Response 2: Thanks for your valuable advice. We revised the sentence to read, “A significantly positive spatial correlation of NGPR exists in Chinese counties, while distribution has a geographical clustering effect. The degree of spatial clustering is also increasing.”

Point3:Also for formulas in Lines 116, 119, 144 and 173 please delete frame.

Response 3: Thanks for your valuable advice. We removed the lines surrounding each equation.

Reviewer 3 Report

The present study mapped the spatiotemporal evolution of NGP in 31 provinces of China. Authors also studied the underlying factors ranging from weather factors, topography, soil factors and socioeconomic factors. The interactive effect of two independent variables were also observed on NGP growth rate. I have four queries/comments:

1. Why interactive effects of only two independent variables were observed? Why not more than two? I believe that agriculture over an given area is a sum effect of all these independent variables acting in harmony in nature. 

2. Authors have not given detailed discussion about the the third research objective: providing a scientific basis for the formulation of NGP....It needs to be discussed.

3. Tables: a) in Table 4, abbreviations such as X1 to X11 need to be given in detail at the footnote; b) in Table 5, why factors X10 and X11 were not studied in interactive detection?

4. The discussion section is too small. The results explained in section 3.1.2., 3.2.1. and 3.2.2. needs discussion.

Author Response

Response to Reviewer 3 Comments

Point 1: Why interactive effects of only two independent variables were observed? Why not more than two? I believe that agriculture over an given area is a sum effect of all these independent variables acting in harmony in nature.

Response 1: Thanks for your valuable advice. In this study, we used geographic detectors to study the driving mechanisms. Geographic detectors are a widely used method in current geographic research. The geographical detector model uses statistical methods to study the spatial heterogeneity of geographical phenomena and identify the factors driving this heterogeneity. Geographic detectors assume that multiple subregions can be distinguished in an area. When the sum of the subregion variances is less than the total regional variance, spatial heterogeneity is present. If the spatial distribution between the two variables is consistent, there is a statistical correlation between the two variables. The core idea is that if an independent variable strongly influences the dependent variable, the spatial distribution of the independent variable and the dependent variable should be similar. This model includes four sub-detectors: factor detection, risk detection, interaction detection, and ecological detection. Our study mainly used factor detection and interaction detection. Before performing interactive detection, factor detection is performed to determine the dominant factor. Then, two-by-two interactive detection is carried out according to the dominant factor. We appreciate your insightful suggestions.

Point 2: Authors have not given detailed discussion about the the third research objective: providing a scientific basis for the formulation of NGP....It needs to be discussed.

 Response 2: Thanks for your valuable advice. We have rewritten the Discussion section of the manuscript to expand the discussion on how NGP is regulated.

Point3: Tables: a) in Table 4, abbreviations such as X1 to X11 need to be given in detail at the footnote; b) in Table 5, why factors X10 and X11 were not studied in interactive detection?

Response 3: Thanks for your valuable advice. We have added a footnote to Table 4 based on your suggestion. Regarding the absence of X10 and X11 in interaction detection (Table 5), we first determined the dominant factors: X1, X2, X3, X5, X8, X9. Then, the five dominant factors were tested for interaction. As X10 and X11 were not dominant factors, they were not tested for interaction.

Point 4: The discussion section is too small. The results explained in section 3.1.2., 3.2.1. and 3.2.2. needs discussion.

Response 4: Thanks for your valuable advice. We have substantially expanded the Discussion section to increase the depth and breadth of our study.
